# Primary Humoral Immune Deficiencies: Overlooked Mimickers of Chronic Immune-Mediated Gastrointestinal Diseases in Adults

**DOI:** 10.3390/ijms21155223

**Published:** 2020-07-23

**Authors:** Ida Judyta Malesza, Michał Malesza, Iwona Krela-Kaźmierczak, Aleksandra Zielińska, Eliana B. Souto, Agnieszka Dobrowolska, Piotr Eder

**Affiliations:** 1Department of Gastroenterology, Dietetics and Internal Diseases, Poznan University of Medical Sciences, 60-355 Poznan, Poland; ida.malesza@gmail.com (I.J.M.); krela@op.pl (I.K.-K.); agdob@ump.edu.pl (A.D.); piotr.eder@op.pl (P.E.); 2Institute of Human Genetics, Polish Academy of Sciences Poznan, 60-479 Poznan, Poland; zielinska-aleksandra@wp.pl; 3Department of Pharmaceutical Technology, Faculty of Pharmacy, University of Coimbra, Pólo das Ciências da Saúde, Azinhaga de Santa Comba, 3000-548 Coimbra, Portugal; souto.eliana@gmail.com; 4CEB—Centre of Biological Engineering, University of Minho, Campus de Gualtar, 4710-057 Braga, Portugal

**Keywords:** primary immunodeficiency, selective IgA deficiency, common variable immunodeficiency, celiac disease, inflammatory bowel disease, Crohn’s disease, ulcerative colitis

## Abstract

In recent years, the incidence of immune-mediated gastrointestinal disorders, including celiac disease (CeD) and inflammatory bowel disease (IBD), is increasingly growing worldwide. This generates a need to elucidate the conditions that may compromise the diagnosis and treatment of such gastrointestinal disorders. It is well established that primary immunodeficiencies (PIDs) exhibit gastrointestinal manifestations and mimic other diseases, including CeD and IBD. PIDs are often considered pediatric ailments, whereas between 25 and 45% of PIDs are diagnosed in adults. The most common PIDs in adults are the selective immunoglobulin A deficiency (SIgAD) and the common variable immunodeficiency (CVID). A trend to autoimmunity occurs, while gastrointestinal disorders are common in both diseases. Besides, the occurrence of CeD and IBD in SIgAD/CVID patients is significantly higher than in the general population. However, some differences concerning diagnostics and management between enteropathy/colitis in PIDs, as compared to idiopathic forms of CeD/IBD, have been described. There is an ongoing discussion whether CeD and IBD in CVID patients should be considered a true CeD and IBD or just CeD-like and IBD-like diseases. This review addresses the current state of the art of the most common primary immunodeficiencies in adults and co-occurring CeD and IBD.

## 1. Introduction

Autoimmune diseases of the gastrointestinal (GI) tract are increasingly growing worldwide over the last decades. They concern both inflammatory bowel disease (IBD) and celiac disease (CeD) [1,2]. This seems to be due to a true rise in incidence rather than increased awareness and detection [3,4,5].

Rising morbidity of both diseases, on the one hand, forces physicians to increase alertness concerning GI symptoms, and on the other hand, it encourages researchers to look for conditions, that can affect diagnostic process and management. The increasing body of evidence that primary immunodeficiency (PID) can complicate diagnostics of CeD [3] and mimic IBD [6] implicates the need for a comprehensive review of this topic, especially when several studies have shown that autoimmune manifestations are the second most common manifestation of PIDs after infections [7,8].

PIDs are usually considered as pediatric ailments and awareness of the problem among paediatricians is relatively high, whereas between 25 and 45% of all PIDs are diagnosed in adulthood [9]. Over the years, an increasing number of diagnoses of PIDs are being made in adults, and recent studies estimate that up to 1:1200 people in the United States are diagnosed with some form of primary immune deficiency [10]. Besides, the vast majority of adult patients with PIDs are not diagnosed or treated early in their course [11], possibly due to a lack of up-to-date knowledge and low awareness of the occurrence of PIDs in adults among physicians [9,12]. Moreover, some researchers suggest that autoimmune disorders are developed in a course of PIDs as patients get older and, for this reason, autoimmunities are more common in adults than in children [9], making this topic even more relevant in terms of CeD and IBD.

According to Agarwal and Mayer, if patients present atypical GI symptoms or are refractory to conventional therapy, the underlying primary immune disorder should be taken into consideration to initiate appropriate treatment [13]. Additionally, a very severe course of disease and need of multiple immunosuppressive agents, or total parenteral nutrition, could be indicative for PID [14]. The recent literature provides an increasing number of publications on IBD related to PIDs; however, they are mainly focused on the child population and concerning very early onset IBD with underlying monogenic diseases [15]. Not much data on IBD related to PIDs in adults can be found.

Among all PIDs, more than 50% make up abnormalities in humoral immunity [16], making immunoglobulin deficiency the most common PID in children and also among adults. In the latter group, selective immunoglobulin A deficiency (SIgAD) and common variable immunodeficiency (CVID) are the most common diagnoses [9].

This review aims to present the up-to-date knowledge on the incidence, pathophysiology, symptoms, diagnostics, and management of autoimmune GI diseases, specifically CeD and IBD, in patients with underlying SIgAD or CVID. Moreover, this review is focused on differences between the classic forms of the above-mentioned diseases and those observed in patients with compromised humoral immunity (as shown in Figure 1), to estimate whether we are facing a spectrum of one disease or different diseases characterized by a similar clinical manifestation.

## 2. Selective Immunoglobulin A Deficiency

### 2.1. SIgAD: Epidemiology and Diagnostic Criteria

SIgAD is the most common PID and, at the same time, the most common immunoglobulin deficiency. Its prevalence varies depending on the ethnicities and regions across the world. It ranges from 1:142 in Caucasians (however, in a great majority of European countries, the estimated occurrence is 1:600) to 1:18 550 in Japanese blood donors [17,18,19]. There were many attempts to establish the prevalence of SIgAD through national and international surveys and registries; however, the true prevalence is certainly underestimated. The main reason is the asymptomatic course of the disease that occurs in 65–75% [20] and up to 85–90% [21] patients, depending on the author. Additionally, different criteria for diagnosing SIgAD were used by researchers in different studies. Since 1999, according to Pan-American Group for Immunodeficiency (PAGID) and European Society for Immunodeficiencies (ESID), SIgAD is diagnosed in an individual older than 4 years with serum IgA level <7 mg/dL but a normal IgG and IgM serum level and with the exclusion of other causes of hypogammaglobulinemia, as well as with the normal response to vaccinations [22]. These criteria have not been changed in the latest ESID recommendations from 2019 (Table 1) [23]. Despite this, some studies used a criterion of the IgA serum level as less than 5 mg/dL [24]. All of this contributes to great difficulties in estimating the real occurrence of SIgAD (presented in Figure 2).

As mentioned above, country of origin and ethnicity affect the prevalence of SIgAD; it is far more common in Caucasians than in Asians [19], as genetic factors likely play a role in the pathogenesis of SIgAD. It occurs far more often in populations with a high percentage of consanguineous marriages; in some studies, even one-third of patients were from consanguineous unions [25]. SIgAD also exhibits familial aggregation, because 20–25% of SIgAD patients have a family history of SIgAD or CVID [26].

### 2.2. SIgAD: Etiology

Up to date, multiple possible mechanisms for this PID have been suggested; however, the exact pathogenesis remains unknown. Many researchers have investigated different mechanisms underlying SIgAD. However, none of them have exhausted the subject on their own, as a complex combination of various causes potentially contributes to SIgAD development, or rather various causes lead to a common clinical manifestation.

SIgAD can be associated not only with an intrinsic B-cell lymphocyte defect but also T-cell dysfunction and impairment in cytokine networks [27,28]. However, a fundamental defect in SIgAD seems to be impaired maturation of IgA-bearing B-cells into IgA-secreting plasma cells [29]. Moreover, studies have proven that IgA-deficient patients have defects in the process of IgA class switching recombination (CSR), production, as well as secretion of IgA, and long-term survival of IgA-switched memory B-cells and plasma cells, possibly due to an increased rate of apoptosis [30,31,32]. The latter leads to a paucity of B-cells secreting IgA both in the plasma and GI tract mucosa, in contrary to a healthy population [33,34]. The defect is possibly present in stem cells, as cases of transfer of SIgAD by bone marrow transplantation have been reported [35].

Many authors suggested a role of impaired T-cell function in the development of SIgAD, as possibly defective antibody production or secretion may be due to dysfunction or decreased activity of different subpopulations of T-cells [30,36]. However, most of the studies concerning the role of T-cells in the pathogenesis of SIgAD are inconclusive. Recent outcomes indicate a defect in regulatory T-cells (Tregs) in SIgAD patients with autoimmunity [28,37]. On the other hand, Lemarquis et al. reported no differences between SIgAD patients and healthy controls, nor in Tregs, in any of the measured T-cell subpopulations (CD3+, CD4+, CD8+, naïve, memory, and differentiated T-cells) [38].

Other important elements that likely account for the development of SIgAD are cytokines and their receptor milieu, as B-cell stimulation via this network is responsible for B-cell differentiation and CSR. Therefore, a combination of abnormalities in the cytokine pattern releasing at local sites of B-cell proliferation, or cytokine kinetics, can contribute to the failure of IgA production in SIgAD patients [30]. This involves transforming growth factor-beta (TGB-β) and downstream transcription factors, which indicates a key role in this process [39]. Another receptor participating in the process is CD40, which stimulates CSR by the T-cell-mediated pathway. T-cell-independent IgA CSR engages other receptors, cytokines, and transcription factors, which includes inter alia toll-like receptor, B-cell receptor (BCR), nitric oxide, retinoic acid, IL−6, TACI (transmembrane activator and CAML interactor), BAFF (B-cell activating factor), A proliferation-inducing ligand (APRIL), and thymic stromal lymphopoietin [30]. Defects in all the aforementioned particles can account for SIgAD. Furthermore, abnormalities in several cytokines, including IL−4, IL−6, IL−7, IL−10, and most newly IL−21, has been found in SIgAD [29,40,41]. This cytokine profile may be changed due to the abnormalities of B-, T-, and dendritic cells (DCs) [30].

It is established that the presence of proper MHC molecules plays an important role in presenting antigen from B-cell to T follicular helper cell (Tfh) and the consequent cytokine production. So, it is not surprising that certain MHC haplotypes are associated with an increased risk of SIgAD. This includes *HLA-B8-DR3-DQ2* (45% of SIgAD patients compared to 16% of the general population, this haplotype predisposes strongly to autoimmunity [42,43]), *HLA-DR7-DQ2*, and *HLA-B14-DR1-DQ5* haplotypes, which are considered strong risk factors for SIgAD, while *HLA-DR15-DQ6* has a protective influence against SIgAD [44,45]. However, there is a suggestion that only familial cases of SIgAD have an HLA genetic-associated background, which is not the case in “sporadic” forms of SIgAD [46].

Except the association with certain HLA haplotypes, the occurrence of SIgAD can also be related to monogenic mutations and cytogenetic abnormalities. The former includes genes involved in multiple different immunity aspects, for example, genes regulating cellular and humoral immunity (e.g., *JAK3*, *DOCK8*, *LRBA*, *DCLRE1C*, *RAG1*, and *CD27*), genes predominantly associated with antibody deficiencies (e.g., *TACI*, *BTK*, *TWEAK*, *MSH6*, *PIK3R1*, *MSH2*, and *CARD11*), and in genes associated with phagocytic defects (e.g., *RAC2*, *CYBB*, *NCF1*, and *SBDS*) [40,47]. Literature indicates a greater number of monogenic mutations, which account for SIgAD; however, this is beyond the scope of this review. Chromosome abnormalities and cytogenetic defects comprise monosomy 4p, trisomy 8, trisomy 10p, translocation of 10q to 4p, 17p11.2 deletions, 18q-syndrome, 18p deletions, trisomy 21, monosomy 22, and 22q11.2 deletion syndrome [48,49,50,51,52,53].

It is suggested that secretory IgA promotes the development of a “healthy” microbiota [54,55]. Besides, alterations in the gut microbiota composition have been described in patients with SIgAD [56]; however, it is difficult to establish whether it contributes to the development of immunoglobulin deficiency, or rather it is a result of impaired function of the mucosal barrier in a condition of the paucity of secretory IgA. Nevertheless, patients with SIgAD are expected to present more severe GI problems, if defects are present in the microbiota [30]. This is possibly due to fact that IgM, which increases compensatorily in the gut mucosa, is less specific than secretory IgA [56].

### 2.3. SIgAD: Symptoms and Clinical Course

As aforementioned, the great majority of patients diagnosed with SIgAD are asymptomatic; however, as expected in PID, some may develop various clinical symptoms. The presentation of SIgAD can be classified according to major symptoms into five phenotypes: Asymptomatic, minor infections, allergies, autoimmunity, and severe [57]. In terms of clinical manifestation, SIgAD patients can also be split into distinct groups: Pulmonary diseases, allergies, autoimmunity, GI disorders, and malignancy [40].

Asymptomatic patients are usually co-incidentally diagnosed during blood donation or routine laboratory evaluation. The mostly asymptomatic course of SIgAD is not fully understood yet; however, it is suggested that it is due to the fact that in this group, a compensatory increase in IgM in the GI mucosa and a rise in the production of IgG occurs [48].

Infections are the most common symptomatic manifestations, more frequently in the upper respiratory tract infections [29]. These are mostly caused by encapsulated bacteria, e.g., *Haemophilus influenzae* and *Streptococcus pneumoniae* [21]. It seems important that infections are more likely to occur and have a more severe course in SIgAD patients with concurrent IgG2 subclass deficiency [58,59].

Allergies are also associated with SIgAD, as it has been estimated that 25–50% of SIgAD patients are recognized by an evaluation of allergic diseases [29]. Among others, an increased occurrence of allergic conjunctivitis, rhinitis, urticaria, atopic eczema, food allergy, and bronchial asthma have been reported in individuals with SIgAD [20,29,60].

As IgA comprises two-thirds of all produced immunoglobulins and has an important function in humoral and mucosal immunity [61], it is understandable why GI symptoms are observed in SIgAD patients. However, even though secretory IgA is the major antibody in the intestinal mucosa, the prevalence of GI disorders in patients with SIgAD seems not as high as it would be expected [62]. Giardiasis, nodular lymphoid hyperplasia, CeD, pernicious anaemia, chronic hepatitis, biliary cirrhosis, and IBD are mentioned most frequently in the literature concerning comorbidity of GI diseases and IgA deficiency [6,40].

One of the most intriguing features of SIgAD is its association with multiple autoimmune diseases. Besides, it has been indicated that between 26% and 32% of patients with SIgAD are affected by an autoimmune condition [25,37,63]. Interestingly, this percentage is lower among the pediatric population, perhaps due to fact that autoimmune disorders require a longer time to develop [57,64,65,66,67], and usually manifest in patients in the second decade of life or later [57]. Autoimmune disorders, which are observed in patients with SIgAD with a higher frequency than among the healthy population, and include, inter alia, idiopathic thrombocytopenic purpura, autoimmune haemolytic anaemia, Graves’ disease, type 1 diabetes mellitus, thyroiditis, CeD, IBD, and systemic lupus erythematosus [68,69]. Few hypotheses explaining the association between SIgAD and autoimmunity have been suggested, which are described thoroughly by Odineal and Gershwin [68]. The first one indicates that certain HLA phenotypes, including phenotype 8.1 (*HLA-B8-DR3-DQ2*), favour the development of both SIgAD and autoimmune diseases [70]. A second hypothesis suggests that the same B-cell, T-cell, or cytokine abnormalities may underline the development of SIgAD and autoimmunity, as a deficiency of Tregs is observed in both [28,37]. However, as discussed above, this theory is controversial, as different authors have obtained contradictory results [28,38,71]. A third proposed mechanism suggests that monogenic mutations can predispose to SIgAD and autoimmune disorders. For example, a similar mutation of *CTLA4-ICOS* is present in SIgAD, CVID, and CeD [72]. Bronson et al. reported a few monogenic associations between IgA deficiency and autoimmune manifestations [73]. Last, but not least, IgA has a key role in the mucosal barrier, as it protects from entering foreign pathogens and antigens. In the condition of lowered secretory IgA, the mucosa possibly has greater permeability. This may cause increased sensitisation of B- and T-cells, and can place patients at an increased risk of autoimmune disorders [20,25,60,64,74,75,76,77]. Besides, it is established that IgA has anti-inflammatory properties, likely via the clearance of pathogens and immune complexes [25] and interactions with certain receptors, such as FcαR (subgroup of Fc receptors). All these contribute to the downregulation of immune pathways, which propagate inflammation [67].

Although malignancies occur more often in many PIDs, likely as an expression of impaired immunological control, it is not established whether it is a case for SIgAD, as data provided by the literature in this topic is not conclusive [78,79,80]. In SIgAD, cases of carcinomas (particularly adenocarcinoma of the stomach) and lymphomas (usually of B-cell origin) have been reported [80,81,82,83]. A graphic summary concerning SIgAD is illustrated below in Figure 2.

## 3. Common Variable Immunodeficiency

### 3.1. CVID: Definition and Epidemiology

CVID is an umbrella name for the most common symptomatic PID of heterogeneous etiology [84,85], with an estimated incidence of 1:10,000 to 1:50,000 in the Middle East and Caucasian population [86,87,88]. It is less frequently described in the African and Asian population, possibly due to differences in the availability of appropriate diagnostics, registry data, and awareness of PID [89]. Similarly to SIgAD, CVID exhibits familial clustering, as about 15% of individuals have first-degree relatives with SIgAD [90], and the occurrence of CVID is increased in populations with a high rate of consanguinity marriages [91,92].

The diagnosis of CVID is usually made between the ages of 20 and 40 and only about 20% are diagnosed under the age of 21, so it can be considered as adult-onset PID [93]. Besides, a diagnostic delay of 6 to 8 years after the first presenting manifestation often occurs [94].

### 3.2. CVID: Diagnostics and Criteria

CVID is considered as a heterogeneous group of PID, characterized primarily by hypogammaglobulinemia, increased susceptibility to infections or other symptoms associated with PID, and impaired responses to infections or vaccinations [95,96]. As CVID has a highly heterogeneous character in terms of both immunological features and clinical manifestations, accurate diagnostic criteria are of utmost importance. However, they vary slightly according to different sources. The International Consensus Document on CVID published in 2016 allows the diagnosis of CVID in patients with IgG levels at least two SD below the age-appropriate reference and decreased IgA or IgM serum levels with poor or absent antibody response to vaccination. The patient must be at least 4 years old and secondary cause of hypogammaglobulinemia has to be excluded—this applies to known genetic defects [86]. The ESID criteria seem to be more complex, as, except the aforementioned immunological features, they also include the presence of symptoms: Increased susceptibility to infections, autoimmune disorders, granulomatous disease or unexplained polyclonal lymphoproliferation, and familial history of antibody deficiency. Moreover, the ESID criteria comprise low switched memory B-cells (<70% of the age-related value) and underline the need to exclude profound T-cell deficiency, and not only secondary hypogammaglobulinemia (Table 2). Besides, depending on how many classes of antibodies are decreased in an individual, ESID classified patients into two categories, probable CVID and possible CVID [95].

### 3.3. CVID: Etiology

A crucial abnormality observed in CVID is hypogammaglobulinemia, so it is not surprising that in CVID patients, lymphocytes B are commonly affected. However, impairment of other elements of the immune system, both adaptive and innate, can also be associated with the development of CVID [84]. Most studies indicate that about 90% of CVID patients have normal B-cell counts [97,98], suggesting that the major defect is possibly related to the differentiation of B-cells into memory and plasma cells [97]. Besides, abnormalities in CSR and somatic hypermutation (SHM) are also present [99,100,101]. Intrinsic defects of B-cell activation leads to disturbances in the production or secretion of antibodies [101]. As in patients with CVID B-cells present an impaired response to BCR and tool-like receptor (TLR) ligation, it seems understandable why these individuals fail to produce and secrete high-affinity class-switched antibodies when exposed to vaccination or infections [100,102]. Furthermore, BCR and TLR have a key role in counterselection of autoreactive B-cells, and defects in their function have a permissive impact on autoimmunity development [101].

Many researchers pointed out the presence of abnormalities in B-cell subsets. These may result from an impaired germinal centre reaction, which is required for the correct development of switched memory B-cells [84,103]. A decrease in IgM memory B-cells (CD19+/CD27+), class-switched memory B-cells (CD19+/CD27+/IgD−/IgM−), and plasma cells have been reported [104,105,106]. This is possibly caused by enhanced terminal B-cell apoptosis [32,107]. López-Gómez et al. reported an elevated level of proapoptotic Bax and Bim proteins, which correlated with increased apoptosis of CD27+ memory B-cells [108]. Another example is the expansion of CD21^low^ B-cells, characterised by low expression of CD21 and CD38 simultaneously [109]. An increased number of CD21^low^ B-cells is related to immune dysregulation, possibly caused by the rise in interferon-gamma (IFN-γ)-producing CD4+ CXCR5+ Tfh cells [110] and can be involved in autoimmune complications in CVID patients [111].

As aforementioned, only in approximately 10% of CVID patients is the level of B-cells is significantly lowered and B-lymphopenia present, which is a rare phenotype of CVID. This is possibly associated with a disturbance in early bone marrow-dependent B-cell development, including arrest at the pre-B-cell stage [101,112], abnormalities in the expression of surface IgM, and disrupted rearrangements of immunoglobulins’ heavy and light chains [101].

Although the ESID criteria require the exclusion of profound T-cell immunodeficiency, and since it is also not clear whether individuals with proven T-cell defects should be considered as CVID affected [113,114], most authors are inclined to favour the theory that abnormalities in T-cell function are common among patients with CVID [115,116]. These abnormalities comprise aberrations in the total numbers, percentages, surface markers, and function of various T-cell subpopulations [115].

In CVID patients, a decreased number of total, naïve, and memory CD4+ T-cells has been observed, while in contrast, increased activated CD4+ has been reported [117]. It has been suggested that a decreased total count might be due to lowered thymic output and enhanced spontaneous CD4+ apoptosis [117], and an increased number of activated CD4+ may result from defects in regulatory B-cells (Bregs) [118]. The same regularity has been reported in terms of CD8+ T-cells. According to numerous studies, the naïve and effector memory CD8+ count was lowered, while higher percentages of activated CD8+ were observed [113,119,120].

The literature concerning T-cell-dependent cytokines in terms of CVID is inconclusive. Some studies indicate significantly increased levels of Th2 cytokines, such as IL−4 and IL−10, simultaneously with an elevated serum level of CD30 (an indicator of Th2 cytokine production) [121], while others report excessive Th1 responses in patients with CVID [122,123]. Cambronero et al. suggested that an increased Th1 response may be associated with aberrations in monocyte function. Reported overexpression of IL−12 can lead to the upregulation of IFN-γ in certain T-cell subsets [124]. However, other studies reported lowered production of IL−12 in CVID patients [125,126], thus this topic requires further investigation.

Besides, a decreased number of circulating Th17 lymphocytes was shown [127]. It is not fully established whether regulatory T-cells play an important role in the pathogenesis of CVID. On the one hand, some studies reported no significant differences in the percentages and absolute count of Tregs between CVID patients and controls [128]. On the other hand, other researchers showed a lowered number of Tregs in CVID patients, contributing to autoimmunity [129,130], as Tregs of CVID patients exhibit the less effective function of suppressing autoreactive CD4+ [131].

Not only lymphocytes can be defective in CVID patients. Some researchers suggested that, in CVID-affected individuals, the total number of DCs might be lowered, and their maturation and stimulatory function might be impaired [125,126]. Defects of DCs, having a key role in presenting antigens to T-cells, may result in decreased generation of antigen-specific CD4+, which can subsequently impair antibody production and secretion [84].

With respect to monocytes/macrophages, as mentioned above, an alternation in the cytokines produced by these cells was observed. Besides, enhanced monocyte/macrophage activation, perhaps due to persistently increased tumour necrosis factor (TNF) levels and increased TNF receptor expression, was reported [132]. Additionally, a 5-fold increase of the monocyte fusion rate was indicated, which could contribute to chronic inflammation and excessive susceptibility to granuloma formation [84].

Innate lymphoid cells (ILCs), including natural killer cells (NK cells), can also be affected in patients with CVID; however, this topic is not well investigated, and the literature provides many contradictory data [84]. Differences within NK cell subsets were reported between CVID patients and healthy controls [133]. It is suggested that this can be a compensatory mechanism for protection against malignancies and viral infections, as an increased occurrence of both conditions is observed in CVID patients with an NK cell deficiency [134].

Multiple monogenic mutations have been associated with CVID development [135]; however, there is an ongoing discussion whether, in the case of the established causative mutation, the condition should be classified as CVID, or perhaps more appropriately, as CVID-like disorder [136].

Nevertheless, among patients fulfilling CVID criteria, monogenic disorders were identified in less than 20% in nonconsanguineous cohorts, and in approximately 70% in cohorts of patients from consanguineous unions [137].

Mutations detected in CVID patients include genes associated with impaired B-cell development (e.g., *IKZF1*, *BAFFR*, *TWEAK*, *CD27*, *STAT1* GOF, *NFKB2*, *IRF2BP2*), impaired CSR/SHM (e.g., *BACH2*, *IL21*, *IL21R*), excessive lymphoproliferation (e.g., *CTLA4*, *LRBA*, *PIK3CD*, *STAT3* GOF), and impaired B-cell activation and tolerance (e.g., *NFKB1*, *TACI*, *CD19*, *CD21*, *ICOS*, *BLK*, *PLCG2*, *CD81*, *CD20*) [88,135,138,139]. It is important to note that many of the genes mentioned above are also involved in SIgAD pathogenesis. The exact pathomechanism underlying immunodeficiency development in the case of the presence of the aforementioned mutations goes beyond the subject of this review; however, up-to-date genetic defects associated with PIDs, including CVID, have been enlisted in the IUIS 2019 classification [140].

CVID is also associated with certain HLA haplotypes, usually common with ones predisposing to SIgAD. These include haplotypes *HLA A1–B8-DR3* and *B14-DR1* [141,142].

Similarly to SIgAD patients, alternations in the gut microbiota have been described in CVID patients. These changes can be associated with a more severe disease phenotype [143], as it is suggested that abnormalities in the gut microbiota can lead to increased mucosal permeability, resulting in increased bacterial translocation, an elevation of lipopolysaccharides, alongside chronic immune activation [144]. Interestingly, a recent study in a mouse model of CVID with CD19 deficiency (B-cell receptor important for B-cell development) revealed anaerobic bacteria outgrowth in the gut that was associated with chronic inflammation of the gut, and resulted in malabsorption [55]. Additionally, similarly to SIgAD, it is not explained whether the described microbiota alteration contributes to CVID development or is an expression of the compromised immune system [145].

### 3.4. CVID: Symptoms and Clinical Course

As CVID is considered a very heterogeneous disorder, appropriate classification of affected patients is essential. Many classifications of CVID patients, based both on clinical manifestations and immunological features as B-cells subsets, have been proposed to this day [146]. Despite this, different authors take a different approach to describing CVID symptoms and classifying them into clinical phenotypes. Some of them also prefer to describe typical CVID manifestations as complications caused by the condition, not a phenotype [84]. Cunningham-Rundles highlighted probably the two most important manifestations of CVID: Infections and autoimmune/inflammatory [147]; however, other researchers underline the importance of lymphoproliferative syndromes, malignancies and enteropathy, asthma, allergies, and chronic pulmonary diseases [84,86,88,148]. Nonetheless, according to Chapel et al., more than 80% of CVID patients present with one of the aforementioned phenotypes/manifestations [148].

As in most PIDs, infectious complications are the most typical manifestation of CVID. These include mainly respiratory tract infections [149], caused usually by *Streptococcus* spp., *Haemophilus* spp., *Moraxella catarrhalis*, *Neisseria meningitidis*, and *Staphylococcus* spp. [150,151], a recurrent or severe course of which may result in complications, such as bronchiectasis and interstitial lung disease [147,152,153]. The latter may be the initial manifestation of CVID in a certain group of patients [154]. GI tract infections are also very common and usually manifest as persistent or acute diarrhoea [155]. *Giardia lamblia*, *Campylobacter jejuni*, and *Salmonella* spp. are the most commonly identified pathogens and are usually associated with a low or undetectable level of IgA [156,157].

Autoimmune manifestations are reported in approximately 30% of CVID patients [148,158,159], among which autoimmune cytopenias are the most frequent and potentially dangerous manifestations [160]. Autoimmune thyroiditis, rheumatoid arthritis, Sjogren syndrome, and multiple other rheumatologic diseases have also been observed [161,162,163,164,165]. It is suggested that aberrations in T-cell subsets, especially Tregs, play an important role in the development of autoimmunity in CVID patients [166,167].

Other important manifestations commonly affecting CVID patients are inflammatory disorders [168], often described together with autoimmunities. In large European and United States cohorts, approximately 70% of CVID patients suffered from one of those [148,160]. Typical for CVID seems the presence of granulomas in various organs. The granulomatous disease occurs in 8–22% of individuals with CVID [147], and these granulomatous changes might be misdiagnosed as sarcoidosis, leading to delays in the recognition of PID [169,170]. Another important form of granulomatous disease uniquely associated with CVID is granulomatous lymphocytic interstitial lung disease [171].

The GI tract is frequently affected in individuals with CVID [93,160,172]. Apart from infections of the GI tract, non-infectious enteropathy occurs in 20% to 60% of CVID patients. This might resemble IBD or CeD [147]. Interestingly, Pensieri et al. indicated that in patients with CVID and villous atrophy, the mortality rate is higher than in CVID individuals without enteropathy [173]. It is suggested that the increased prevalence of GI symptoms among CVID patients in contrast to other antibody deficiency syndromes is due to T-cell dysfunction. The fact that immunoglobulin replacement therapy does not alleviate enteropathy symptoms is in favour of this theory [174].

Last, but not least, patients with CVID have a predilection to develop lymphoproliferative syndromes (e.g., persistent lymphadenopathy, lymphoid interstitial pneumonia, hepatomegaly, splenomegaly), and malignancies, which include primarily lymphomas [86,148,175]. CVID patients have an estimated 30-fold increased risk of developing the latter [83,176]. The most common malignancy is non-Hodgkin lymphoma; however, CVID patients also have a significantly increased risk for other malignancies, such as colorectal cancer, breast cancer, uterine cancer, and neurogenic malignancies [147].

## 4. SIgAD Progression to CVID

Intriguingly, the literature provides information concerning the progression of SIgAD to CVID [177]. It is suggested that it could be due to the similar pathogenesis of both entities. For example, the same HLA haplotypes, including *HLA A1–B8-DR3* and *B14-DR1*, predispose to both [147]. Additionally, familial clustering of SIgAD and CVID is in favour of this theory [178,179,180]. Interestingly, one study indicated that more than 20% of patients with SIgAD and autoimmunity progressed to CVID, which is not a case for SIgAD patients without autoimmune disorders [25]. Despite CVID diagnostic criteria including a lack of antibody response to vaccination, it seems that some CVID patients can produce them in detectable titers [25,177]. This could be proof of a progression from SIgAD to CVID and reflect the transition stage. Finally, it still raises doubts whether these patients had true SIgAD progressing to CVID or suffered from subclinical or underdiagnosed CVID [68].

## 5. Celiac Disease

### 5.1. CeD: Definition and Epidemiology

CeD is an autoimmune disorder characterized by the presence of small-intestinal enteropathy triggered by gluten ingestion (specifically, gliadin peptides that can be found in wheat, rye, and barley) in genetically predisposed individuals. CeD prevalence is estimated at around 1% of the worldwide population, thus making it one of the most common autoimmune disorders [181,182]. Besides, the incidence of CeD is higher in first-degree relatives of patients with CeD (Singh et al. established CeD prevalence in first-degree relatives at 7.5% and in siblings at 8.9% [183]), and in patients from risk groups, e.g., with Down syndrome, type 1 diabetes, or IgA deficiency [68,184,185,186]. Reports indicate that CeD prevalence varies widely among regions and populations, probably due to varying wheat intake and frequency of *HLA-DQ2*, *HLA-DQ8*, or *HLA-DQ2.5*. *HLA-DQ2* is present in about 20–30% of the general population; however, only 1–3% of all individuals having *HLA-DQ2* develop CeD. Slightly less than 20% of the general population is carrying *HLA-DQ8*, but it is present in only 0.1–0.3% of individuals with CeD [187]. Thus, thess specific HLA haplotypes predispose to CeD development, but they are not causative.

Additionally, CeD diagnosis is more common in females (0.6% > 0.4%) [182]. An increasing prevalence in Asian countries is probably caused by progressive “westernization”, which results in increased wheat consumption [3,188,189]. Studies show an increasing global prevalence of CeD [2,190,191]; however, researchers do not agree whether it reflects a true increase in incidence or progress in CeD diagnostics and increased awareness among physicians [3,192].

### 5.2. CeD: Diagnostics

The diagnosis of CeD is made based on serologic tests together with histological findings, and a positive response to a gluten-free diet (GFD) [193]. The serological examination includes measurement of the anti-transglutaminase-2 (TG2) IgA level in serum, considered as a first-line screening test. In the case of weakly expressed anti-TG2 IgA or patients with an impaired immunological status, anti-TG2 IgG, endomysial antibodies (EMA), or deamidated gliadin peptides (DGP) IgA and IgG levels should be tested [194].

Duodenum endoscopic examination in CeD usually reveals bulb atrophy with visible submucosal vessels, loss or reduction of folds, a mosaic pattern, and mucosal fissuring. Although these signs may occur alone or in combination, one-third of new cases of CeD have a normal endoscopic appearance [195]. Because of this, at least four biopsy samples from the second part of the duodenum should be taken during endoscopy when CeD is suspected [194].

CeD-specific histopathological features in small bowel biopsy specimens are required to confirm the diagnosis in adults [194]. In children, a non-biopsy approach is possible, which is widely described in the European Society Paediatric Gastroenterology, Hepatology, and Nutrition (ESPGHAN) guidelines for diagnosing CeD 2019 [196].

Typical CeD histological findings include villous atrophy, crypt hyperplasia, and an increased intraepithelial lymphocyte (IEL) count. Histopathological changes are graded in the modified Marsh–Oberhuber classification [194,197]. According to the European Society for the Study of Coeliac Disease (ESsCD) guidelines, specimens should be taken after at least 6–8 weeks of a diet containing a minimum of 10 g of gluten [194].

HLA testing is not a part of the routine diagnostic process. It should be performed in doubtful cases, i.e., when serology is negative but histology is strongly suggestive of CeD. It is also useful to exclude the possibility of CeD, especially when GFD is already administered, or to identify patients who require further monitoring among those in a risk group (e.g., family members of patients with CeD, affected with autoimmune diseases, or genetic disorders associated with CeD). This approach is justified by the fact that a negative test for *HLA-DQ2* and *-DQ8* has a positive predictive value >99%, thus helping to exclude CeD diagnosis [194].

### 5.3. CeD: Symptoms

Classic symptoms of CeD are chronic diarrhoea; symptoms of malabsorption, such as weight loss; failure to thrive in childhood; delayed puberty; and short stature; however, they are not very common. This can be confirmed by the fact that, for instance, more than 10% of CeD patients are obese [193]. Symptoms considered as non-classical, including iron deficiency, bloating, constipation, chronic fatigue, headache, abdominal pain, and osteoporosis [3], are more common and present in more than 50% of CeD patients [198].

## 6. Celiac Disease in Selective IgA Deficiency

### 6.1. CeD in SIgAD: Epidemiology

The real occurrence of CeD among patients with SIgAD and SIgAD among patients with CeD is difficult to estimate and varies depending on the author. The difficulties come from usually small study groups, different SIgAD criteria adopted by researchers, and the fact that in many studies, symptomatic SIgAD patients were examined, while the course of the disease is mostly asymptomatic.

Various sources seem to agree on the occurrence of SIgAD among CeD patients, which is approximately 2–3%. Such a value is provided by the ESsCD guidelines for 2020 and is based on a large European cohort study by McGowan et al. [194,199]. Pallav et al. reported an incidence of SIgAD among patients diagnosed with CeD as 1.9% in the United States [200]. Wang et al. indicated a 2.56% prevalence of SIgAD in individuals with CeD [201]. Both cited results agree with the one published in the ESsCD guidelines. Interestingly, in a comprehensive review from 2020, Odineal and Gershwin reported a weighted average of SIgAD in CeD occurrence, based on 11 studies, as 0.57%. However, they indicated that this result might be biased by the aforementioned difficulties and lack of controls in certain studies [68].

The situation is different in terms of the occurrence of CeD among patients diagnosed with SIgAD, as sources seem inconclusive. In 2014, in a large review paper, Singh et al. reported a 10–30% prevalence of CeD in individuals with SIgAD [19]. Original papers also presented a large spread of results. Ludvigsson et al. indicated that CeD occurs in 6.7% patients with SIgAD [202]; in the research of Lenhardt et al., it was 8.7% [203], while McGowan et al. reported 16.6% [199], and Wang et al. showed the result of 15.2% [204]. All cited studies are in favour of the theory that the prevalence of CeD in patients with SIgAD is higher than in the general population; however, this requires further investigation.

### 6.2. CeD in SIgAD: Etiology

Several hypotheses have been formulated to explain the association between SIgAD and CeD. Both diseases share the same predisposing HLA haplotypes [204], described in detail above. The common genetic background partially explains the overlap between CeD and SIgAD. However, an analysis of the literature performed by Wang et al. has driven researchers to the conclusion that the distribution of HLA-DQ haplotypes is similar in CeD patients with SIgAD and IgA-sufficient CeD patients [201]. This indicates that more mechanisms must be underlying the association between SIgAD and CeD. For instance, it might be associated with variants of *CTLA−4* and *ICOS*, since it is established that SIgAD and CeD patients can share a common mutation of *CTLA4-ICOS* [72].

Another hypothesis states that an increased prevalence of CeD among SIgAD patients is an expression of the aforementioned general predisposition to autoimmunity in SIgAD-affected individuals [25,37]. Pallav et al. found that the mean ages at diagnosis in CeD patients with additional autoimmune disorders were higher than those with CeD only [200]. The authors suggested it might be due to prolonged gluten exposure, resulting in ongoing inflammation.

This leads to the next theory. It is suggested that secretory IgA in the gut plays an important role in neutralizing gluten peptide; therefore, it is recommended to delay gluten intake by infants until 4 months of age [205], as low secretory IgA levels in the intestine are normal for all babies [206]. The same mechanism might underline the susceptibility of SIgAD patients to CeD development, as in the absence of IgA, individuals’ mucosa can be more exposed to gluten, which can lead to abnormal processing of these antigens [62].

### 6.3. CeD in SIgAD: Diagnostic Difficulties

The basic principles of CeD diagnostics in SIgAD patients are similar to those applied to IgA-sufficient individuals. The most important difference concerns serological diagnostics. In immunocompetent patients, first-line testing comprises of measurement of the anti-TG2 IgA level in serum [194]. For obvious reasons, this approach is not effective in SIgAD patients, as false negative results might be obtained; thus, patients with SIgAD may elude CeD diagnosis [200,207,208]. To avoid such a situation, ESsCD recommended assessing total IgA levels concurrently with serology testing to determine whether IgA levels are sufficient [194]. In the case of a lowered IgA serum level, IgG anti-TG2 and IgG-DGP are regarded as the best tool for identifying CeD in patients [194,209]. Interestingly, Pallav et al. reported cases concerning SIgAD patients with a positive result of anti-TG2 IgA at diagnosis [200]. The authors hypothesized that in some individuals with SIgAD, gluten-related immune activation can lead to detectable serum IgA levels [210]. However, this should be taken as an exception and should not affect management.

The next step in CeD diagnostics, after serological examination, is a small intestinal biopsy and histological assessment of the taken specimens. Histopathology findings in SIgAD patients with CeD do not vary much from those of patients with CeD alone [48,210]. It is indistinguishable from the pathology seen in non-SIgAD patients with CeD in terms of increased numbers of intraepithelial lymphocytes, villous atrophy, crypt hyperplasia, and infiltration of the lamina propria with lymphoid cells. The main difference concerns IgA-secreting plasma cells, which are absent in CeD patients with coexisting SIgAD [211,212,213].

### 6.4. CeD in SIgAD: Symptoms

Symptoms of CeD in IgA-deficient patients are similar to those in patients with CeD alone, while differences might concern the frequency of occurrence of certain symptoms [6]. On the one hand, Chow et al. and Heneghan et al. reported a higher prevalence of diarrhoea in SIgAD-CeD patients; however, this finding was not significant [214,215]. On the other hand, literature investigation drove Odineal and Gershwin to a conclusion that individuals with SIgAD and CeD are less likely to have GI symptoms [68], and the results of Pallav et al. favour this thesis, indicating a lack of classic GI symptoms in 66.7% of SIgAD-CeD patients compared to 17.7% in patients with CeD alone. However, this finding was also not significant [200]. Further investigation seems required in this field.

In SIgAD-CeD patients, co-existing autoimmune diseases are remarkably more common than in IgA-sufficient CeD patients. Pallav et al. reported the presence of another autoimmunity in 67% of SIgAD-CeD compared to 23.5% of the only CeD control [200], while Chow et al. indicated the second autoimmunity in 29% of antibody-deficient patients compared with 12% of CeD patients with normal IgA levels [215]. Besides, according to Pallav et al., the average age at diagnosis was higher for individuals with CeD and SIgAD than for CeD only [200]. Although this finding was not significant, it remains conclusive with the previously mentioned theory, in that antibody-deficient patients develop autoimmunity later in the course of the disease [57].

### 6.5. CeD in SIgAD: Treatment

As CeD in SIgAD does not differ significantly from CeD in immunocompetent patients, recommended treatment does not vary either. SIgAD patients with CeD are responsive to gluten withdrawal and GFD is recommended [29,62,157,216]. If SIgAD individuals with CeD fail to respond to GFD, it is an indication to broaden the diagnostics and consider other causes of villous atrophy, including giardiasis, small-bowel bacterial overgrowth, or CVID [62,156,194]. Interestingly, persistent elevation of anti-TG2 IgG and IgG-DGP was frequently found in SIgAD adults despite adhering to GFD [204]. Korponay-Szabo et al. reported a very slow decrease of anti-TG2 antibody levels in IgA-deficient patients; most of them were still positive after more than two or three years on a GFD [217], while in non-SIgAD patients, serological normalization is expected within one year of GFD [216,218]. Besides, Chow et al. reported a lack of normalization of serological tests after a long period of GFD (mean 7.25 years) in half of their SIgAD-CeD patients [215]. This might complicate the monitoring of SIgAD-CED patients and implicate the need to intensify endoscopic examination in comparison with individuals with CeD only. It is not established whether it is due to a lack of adherence to GFD, or part of the immune-regulatory defect seen in IgA, or finally, it is associated with a certain HLA haplotype [204].

## 7. Celiac Disease in Common Variable Immunodeficiency

### 7.1. CeD in CVID: Epidemiology

Despite the fact that CVID and SIgAD share a common pathogenesis, in terms of CeD occurrence, these entities are not alike. GI manifestation is very common among CVID patients, and often presents as non-infectious enteropathy, which might resemble other GI diseases like IBD and CeD [93]. However, the prevalence of enteropathy in CVID patients varies depending on the author. In Uzzan et al.’s review, it is reported as 10% to 12%, Yuan and Bousvaros indicated it is between 10% and 20%, but Cunningham-Rundles suggested it might be even up to 20% to 60% of CVID patients [93,147,219]. These values concern the incidence of non-infectious enteropathy, which is not synonymous with CeD. The true prevalence of the latter is very difficult to estimate, as it is not fully established whether we are facing CeD or only celiac-like enteropathy in CVID patients. Some authors even highlighted that the association between CVID and CeD remains controversial [220]. This is discussed in detail below. In 2019, Schwimmer et al. stated that both CeD and celiac-like conditions are common in patients with SIgAD deficiency and CVID [8].

Regardless of the way we classify this entity, CeD-resembling findings, including villus atrophy and increased intraepithelial lymphocytes, are common among CVID patients. According to Malamut et al., villous atrophy was observed in over 50% of CVID patients [156], while Luizi et al. indicated it to be 31.2% [221]. A literature review performed by Hartono et al. reported the presence of villus atrophy in CVID patients within the range of 24% to 50% [222]. However, Karaca et al. showed a lower incidence of celiac-resembling enteropathy in CVID, as it affected only 10% of the study participants [92]. Furthermore in Venhoff et al.’s study, villus atrophy and increased IELs were observed in only 8% of CVID patients presenting with enteropathy [223].

Besides, Giorgio et al. underlined that although celiac-like histological findings can be found in 30% of CVID individuals, less than 10% of patients can be diagnosed with true CeD [224].

### 7.2. CeD in CVID: Etiology

Despite the association between CVID and CeD being controversial and the true prevalence of such a linkage being unestablished, some data concerning a common CVID and CeD pathogenesis seems worth mentioning.

On the one hand, it is established that CVID and CeD can share a common dysregulation of *CTLA−4* and *ICOS* [72]. Additionally, the coexistence of CVID and CeD in the same family was described [225]. Besides, an altered expression of HLA-DR on antigen-presenting cells (APCs) both in CVID and CeD was demonstrated by Viallard et al. [226]. Giorgio et al. suggested that defective antigen presentation by APC through the HLA complex might be associated with the induction of an immunological response shared by CVID and CeD [224].

On the other hand, Jorgenssen et al. assessed CVID patients with increased IEL in the pars descendens of the duodenum and CeD patients by gene expression microarray analyses and HLA typing, and the results indicated a large difference in gene expression, suggesting little or no overlap of CVID and CeD [227]. Moreover, Woodward et al. reported multiple cases of persistent Norovirus infection in CVID patients who presented with celiac lesions, which healed after Norovirus eradication [228]. It requires further investigation to establish whether CVID and CeD are truly associated or share a common etiology.

### 7.3. CeD in CVID: Diagnostic Difficulties

Enteropathy accompanied by malabsorption is a common clinical presentation of CVID; however, villus atrophy remains a challenging finding in this group. Serological diagnostics is insufficient in individuals with CVID, and for this reason, it can resemble seronegative CeD [55,157]. However, in the vast majority of cases, CVID enteropathy differs from CeD not only the results of the serological assays, so it is debated whether it can be classified as true CeD or rather should be called celiac like [156].

Similarly, as in SIgAD, histological assessment of a duodenal biopsy shows a paucity or absence of plasma cells [62]. Besides, villous atrophy is usually less severe and the IEL count is often lower than in CeD, excessive neutrophil infiltration can be observed, as well as graft versus host-like lesions or follicular lymphoid hyperplasia [93,156]. ESsCD guidelines from 2019 say directly that an absence of plasma cells suggests CVID [194].

Another argument in favour of the theory that enteropathy presented in CVID often differs from CeD is a fact that approximately 50% of patients do not respond to GFD [229]. Some sources indicate that this percentage might be even up to 80% [93].

Clear guidelines for the management of CVID patients with enteropathy have not been established yet. Some authors recommend initiating GFD after excluding infectious causes since the first proposed diagnosis is generally CeD. Jorgensen et al. suggested that the only criterion to confirm CeD diagnosis in CVID patients is the histological response to a GFD [230,231]. Other researchers highlighted the importance of HLA determination, as the presence of the *DQ2* or *DQ8* haplotype can be associated with concomitant CeD, thus allowing the identification of CVID patients at risk [223]. Furthermore, if both *HLA-DQ2* and *HLA-DQ8* are negative, the diagnosis of CeD is highly unlikely [194]. Intriguingly, Malamut et al. reported that among 10 CVID patients, in which GFD turned out to be inefficient, 2 were positive for anti-gliadin IgG and 3 patients had the *HLA-DQ2.5* haplotype [156]. This strongly suggests that immune reactivity to gluten cannot alone account for the intestinal lesions in described patients.

There is a growing body of evidence that immunohistochemical analysis of TCRγδ can help discriminate CeD from other causes of duodenal lymphocytosis [224,232]; however, no author suggests the use of this method in the standard diagnostic procedure.

There is no doubt that true CeD can be diagnosed in CVID patients, but yet according to current knowledge, this relationship is rare. Diagnosing enteropathy in patients with CVID requires cautiousness, as data available are scarce and conflicting.

### 7.4. CeD in CVID: Symptoms

Regardless of the cause of symptoms, non-infectious enteropathy characterized by weight loss, abdominal pain, bloating, and severe diarrhoea was reported often in CVID patients [6]. According to Cunningham-Rundles, it was observed in even up to 20% to 60% of CVID patients [147], but Yuan et al. reported a prevalence of the aforementioned symptoms of 10% to 20% [219]. In the vast literature research, we found no data on differences between symptoms of CVID enteropathy and symptoms of CeD, which led us to the conclusion that manifestations of these entities are indistinguishable based on clinical signs.

### 7.5. CeD in CVID: Treatment

Accepted treatment of immunoglobulin deficiency syndromes is immunoglobulin replacement therapy; however, intravenous immunoglobulins turned out to inefficient in CVID-related enteropathy and have not led to a consistent improvement of GI symptoms [156,157].

GFD is recommended only in *HLA-DQ2* or *HLA-DQ8* carriers as an attempt of a 6- to 12-month trial with follow-up histologic assessment [8,93,156]. Patients without *HLA-DQ2* or *HLA-DQ8* or unresponsive to GFD should be treated as a distinct disorder. In this case, short-term steroids or immunomodulators, including azathioprine and 6-mercaptopurine, can be used [62]. There is also evidence for the efficacy of an elemental diet, while in the case of severe malabsorption, limited use of total parenteral nutrition might be required [19,62].

A graphical summary concerning CeD in the discussed PIDs is presented in Figure 3.

## 8. Inflammatory Bowel Disease

IBD is a complex heterogeneous disorder, characterized by chronic idiopathic inflammation of the GI tract. It is associated with dysregulation of the immune system in genetically susceptible individuals in response to environmental triggers [233]. The term IBD refers to ulcerative colitis (UC) and Crohn’s disease (CD). Almost 10% of IBD stays undefined and falls into the category of indeterminate colitis. To make a proper diagnosis and distinguish one from another, clinical, endoscopic, histologic, and radiologic features are used [234,235]. The occurrence of IBD is rising worldwide, with the greatest increases in industrialized regions, which suggests this may be due to increased antibiotic use, decreased exposure to infectious factors, and changes in dietary pattern, including increased intake of emulsifiers and surfactants [236,237,238,239].

### 8.1. Ulcerative Colitis

#### 8.1.1. UC: Definition and Epidemiology

UC is a chronic recurrent inflammatory disorder characterized by continuous inflammation limited to the colon and rectum, with associated bloody diarrhoea and abdominal pain [8,240]. The estimated incidence and prevalence of UC varies depending on the country. For instance, they are higher in Europe and North America and significantly lower in Asia. The prevalence is reported to be 505 per 100,000 in Europe and 286 per 100,000 in the United States, while Asian studies report a much lower prevalence, e.g., 57.3 per 100,000 in Japan and 6.67 per 100,000 in Malaysia [236]. UC usually affects adults aged 30–40 with no sex predominance and leads to disability [5].

#### 8.1.2. UC: Etiology

The exact pathogenesis of UC remains elusive but seems to be complex and multifactorial. Factors contributing to UC development include genetics, host immunity, infections, drugs, gut microbiota, and environmental factors. All this seems to take part in altering the state of intestinal homeostasis in susceptible individuals [8].

Multiple genetic risk factors are likely associated with UC development. However, only 7.5% of these cases are explained by a specific genetic factor [241]. To date, more than 200 loci have been associated with IBD. Among them, *HLA-DQA1* variants are most strongly associated with UC [242]. Except this, mutations in genes related to pathways involved in epithelial barrier function (e.g., *CHD1* and *LAMB1*), cytokine milieu (e.g., *IL1R2*, *IL8Ra/RB*, and *IL7R*), and modulation of inflammation (e.g., *TNFRSF15*, *TNFRSF9*, *PPAR-γ*, *XBP1*) increases the risk of UC [5,243]. In favour of the theory of genetics factors predisposing to UC development is the fact that 8% to 14% of UC patients have a family history of IBD [236,244]. However, all the aforementioned genetic factors are of limited clinical use, since they have little predictive capacity for phenotypes [241,245].

It is suggested that alterations in the permeability of the intestinal mucosa play an important role in triggering UC development. For this reason, enteric infections are suspected as a possible risk factor, as they impair the intestinal barrier [246,247,248,249]. Furthermore, the intake of various drugs can possibly be implicated in the pathogenesis of UC. Among them, antibiotics, nonsteroidal anti-inflammatory drugs, and oral contraceptives are often mentioned [5,250,251]. Interestingly, tobacco smoking is considered as a protective factor since active smokers have a lower risk of developing UC and have a milder disease course in contrast to non-smokers; however, discontinuation of tobacco is one of the strongest risk factors associated with UC [5].

#### 8.1.3. UC: Symptoms

UC is a disease typically presenting with periods of relapse and remission, which are observed in up to 90% of patients. Besides, early relapse is usually associated with a more severe course of the disease [246,252].

The most common symptoms of UC are bloody diarrhoea associated with abdominal pain. Nocturnal diarrhoea, mucous discharge in the stool, urgency, or tenesmus can also be present. Weight loss with anaemia, iron deficiency and hypoalbuminemia, fevers, or even toxic megacolon might be observed in cases of severe, acute, or prolonged untreated UC [5,242]. Extraintestinal symptoms occur in approximately 31% of patients and concern skin, joints, eyes, and liver. Among them, arthropathies are most common [253].

#### 8.1.4. UC: Diagnostics

UC diagnosis is made based on clinical symptoms and endoscopic findings. Histological assessment and serological tests have only an auxiliary function [240].

In patients suspected of UC, colonoscopy with intubation of the ileum and biopsies of affected and unaffected areas should be performed. This approach enables assessment of the extent of the disease and helps to exclude distal ileal involvement, which can be indicative for CD [254].

Endoscopy usually reveals continuous inflammation of the colon, which typically starts in the rectum and can expand on more proximal segments of the large intestine [242]. Characteristic features of lesions include erythema, loss of vascular markings, granularity and friability of the mucosa, erosions, ulcerations, and spontaneous bleeding, with a distinct demarcation between inflamed and non-inflamed mucosa [254,255,256].

Histopathological assessment is useful in defining the extent of disease, as histologic extension can be found even in endoscopically normal-appearing mucosa, and to identify the presence of intraepithelial dysplasia, neoplasia, or cancer [254,257]. Most often, pathologic examination shows inflammation restricted to the mucosa of the colon with distortion of the crypt architecture, crypt abscesses, and infiltration with lymphocytes, plasma cells, and granulocytes, as well as mucous depletion and a lack of granulomas [240,242,257].

Neither serological tests nor imaging studies play a significant role in making the initial diagnosis. Perinuclear antineutrophil cytoplasmic antibodies (pANCAs) can be found in up to 70% of individuals with UC; however, the accuracy of antibody testing for the diagnosis of UC is low. Therefore, it is not recommended as a diagnostic tool [254,258]. Instead, tests for faecal calprotectin, a protein found in polymorphonuclear leukocytes, can be used since a low calprotectin level has a high negative predictive value for IBD [259,260].

#### 8.1.5. UC: Treatment

UC treatment aims to induce and subsequently maintain steroid-free clinical remission. Interestingly, there is no fully agreed or validated definition of remission. The British Society of Gastroenterology (BSG) in recent guidelines proposed the consideration of remission as a combination of symptoms’ resolution (normalization of bowel movements and cessation of bleeding) with mucosal healing in an endoscopy [240].

The choice of therapy varies depending on the severity and localization of the disease. Aminosalicylates, like mesalamine, are used in the treatment of mild to moderate UC [240]. Topical and systemic steroids can be used in flare-ups, while in moderate to severe disease, immunosuppressants and biological treatment are recommended [5,240]. Despite pharmacological treatment, approximately 15% of patients require colectomy [5,242].

### 8.2. Crohn’s Disease

#### 8.2.1. CD: Definition and Epidemiology

CD is a chronic inflammatory disease that affects all segments of the GI tract and is characterized by a relapsing and remitting manner. CD can result in progressive bowel damage and disability [4,261]. There is still a lack of a single unifying definition of CD and the diagnosis itself can remain challenging as it is a combination of clinical, endoscopic, and histopathologic examination [240].

Similarly to UC, CD occurrence is higher in developed countries than in developing countries, and in urban areas than in rural areas [1]. The prevalence of CD is highest in Europe (322 per 100,000) and North America (in Canada and the United States, it is 319 per 100,000 and 214 per 100,000, respectively). The annual incidence in the aforementioned areas is between 10 to 30 per 100,000 [4,236]. In Asia, it is rising; however, it remains significantly lower than in the west, as the crude annual overall incidence is 1.37 per 100,000 [262]. CD presents no sex-specific distribution. It can affect individuals at any age; however, the onset usually occurs in the second to fourth decade of life [4,263].

#### 8.2.2. CD: Etiology

CD is a disease of complex pathogenesis. The multifactorial etiology seems to include a genetic susceptibility, environmental factors, and altered gut microbiota, all of which contribute to an impaired mucosal immune response and compromised epithelial barrier function [4].

Genetic factors predispose to CD development, but they are not sufficient to cause it, as the concordance rates of developing CD in monozygotic twins are approximately 20–50% [264] and about 12% of patients have a family history of CD [244]. Among more than 200 alleles associated with IBD, approximately 40 are specific for CD [241,265]. These include genes associated with innate immunity, preventing the dissemination of invasive bacterial species, related to Th17-cell function and dysregulated cytokine production (e.g., *NOD2*, *ATG16L1*, *Il23R*, *LRRK2*, *IRGM*, *STAT3*, *HLA*, *JAK2*, and Th17 pathways) and impaired mucous production and function (*MUC2*) [241,263,266]. Abnormalities in intestinal tight junctions are also related to IBD [267]. Despite proven participation of genetic factors in the pathogenesis of CD, their usage is still limited to the research field [263].

Tobacco smoking is a well-documented environmental factor. Smoking increases the risk of CD two times [268]. Additionally, drugs can contribute to CD development. It is suggested that antibiotics, by altering the gut microbiota and influencing the intestinal barrier, are implicated in CD pathogenesis [269]. Similarly, as in UC, nonsteroidal anti-inflammatory drugs and oral contraceptives are possibly associated with CD [270]. Interestingly, statins seem to have a protective effect for CD development [271]. Moreover, dietary factors, like a decrease in dietary fibre intake, an increase in saturated fat consumption, and emulsifiers affecting the gut mucosa layer, have been linked with an increased risk of CD [239,272].

It was indicated that IBD patients have a reduced diversity in their gut microbiota when compared to healthy individuals. This is more strongly pronounced in CD than in UC [273,274]. This alternation in the intestinal microbiome can be related to early exposure to pets and farm animals, breastfeeding, hygiene, stress, and diet [275,276,277]. It is not established whether the latter contributes to an increased risk of CD directly, or rather by affecting the gut microbiota [263].

#### 8.2.3. CD: Symptoms

As CD can manifest in any part of the GI tract from the mouth to the perianal area, the symptoms of this disorder are diverse and depend on its location in the gut [4]. In total, 25% of patients present with colitis only, another 25% has ileitis exclusively, and 50% present with ileocolitis. Moreover, 30% of patients suffer from perianal lesions and less than 15% have involvement of the oral or gastroduodenal area [263]. The most common symptom of CD is diarrhoea, often associated with abdominal pain. Diarrhoea can occur due to decreased water absorption and increased secretion of electrolytes, small intestinal bacterial overgrowth, or impaired bile acid resorption [263,278]. Ileocolitis might mimic appendicitis and presents with right lower quadrant abdominal pain and fever. Fatigue, anorexia, and weight loss are common symptoms. Signs of colonic involvement resemble those of UC and might include rectal bleeding or bloody diarrhoea [4,278]. Up to 50% of patients have intestinal complications, including strictures, abscess, fistulas, or phlegmon [278]. Extraintestinal manifestations also occur in 43% of CD patients and include large joint arthritis, uveitis, iritis, episcleritis, erythema nodosum, and pyoderma gangrenosum [4,253,263].

#### 8.2.4. CD: Diagnostics

According to the American College of Gastroenterology (ACG), CD is diagnosed clinically and there are no truly pathognomonic features of the disease. The diagnosis is made based on the results of the endoscopic, radiographic, and histologic assessment with evidence of chronic intestinal inflammation [278].

Ileocolonoscopy with biopsy is considered the first-line investigation for individuals suspected of CD [240]. Endoscopy typically reveals mucosal nodularity, edema, ulcerations, friability, and stenosis. Segmental inflammation (so-called “skip lesions”), apthoid, longitudinal, and serpiginous ulcerations are characteristic in CD. A combination of the aforementioned lesions creates a so-called cobblestone pattern [4,279,280]. Classic endoscopy is insufficient in approximately 20% of patients [278]. Small-bowel capsule endoscopy is recommended in this group [4]. Additionally, computed tomography enterography and magnetic resonance enterography are useful and equally efficient in assessing the small intestine [240,278].

Histological findings typical of CD include chronic focal, patchy, discontinuous, and transmural inflammatory infiltration, and goblet cell preservation [4]. The histological hallmark of CD is the epithelioid granuloma; however, they are seen only in approximately one-third of CD patients [278].

As mentioned above, the routine use of genetic testing to establish the diagnosis of CD is not recommended. Even though more than half of patients might have anti-*Saccharomyces cerevisiae* antibody IgA (ASCA) in plasma, serological markers are of limited use [278,281]. Testing for fecal calprotectin can be useful since the low concentration of calprotectin in the stool has a high negative predictive value for the diagnosis of of IBD [4,282]. C-reactive protein is also useful, as its serum concentration changes dynamically, which makes it a good marker to detect and monitor inflammation [240,278].

#### 8.2.5. CD: Treatment

The choice of medical therapy in patients with CD depends on the disease location and its severity. The goal is to induce and maintain symptomatic remission with signs of improvement of objective indicators of mucosal inflammation and to prevent the development of disease complications, such as strictures and fistulas [278].

Induction of clinical remission is typically obtained by the use of corticosteroids, the administration of which should be limited in time and not exceed 3 months without attempting to introduce corticosteroid-sparing agents [278]. Except steroids, biological treatment, including TNF inhibitors (e.g., infliximab, adalimumab, certolizumab), are effective in inducing remission [240,278]. Steroids are not effective in the maintenance of remission. For this purpose, immunomodulators like thiopurines (e.g., azathioprine, 6-mercaptopurine) and biological treatment are used. Commonly used in UC, 5-aminosalicylic acid has an uncertain effectiveness in induction or maintenance of remission in CD, thus it is not recommended [261,263,278].

Surgical interventions, including bowel resection, stricturoplasty, or drainage of an abscess, are necessary in the vast majority of CD patients during their lifetime; however, surgery is not curative and relapses are observed frequently. Therefore, the main assumption of the surgical approach should be preserving as much of the intestine as possible, since the vast majority of patients require reoperations in the disease course [283,284].

## 9. Inflammatory Bowel Disease in Selective IgA Deficiency

### 9.1. IBD in SIgAD: Epidemiology

Despite the fact that IBD is often mentioned as a common manifestation of PIDs, the real statistics in this topic are scarce. Reliable data concerning monogenic PIDs presenting with an IBD-like phenotype are available. However, not much information about IBD in SIgAD patients can be found and the presented data mostly come from case reports. The first such description comes from 1975 and concerns the coincidence of SIgAD, CeD, and UC in one female patient [285].

Various authors seem to agree that the prevalence of IBD in SIgAD patients is increased in contrast to non-SIgAD patients. Ludvigsson et al., in a large population-based matched cohort study, reported a prevalence of 3.9% among SIgAD individuals to 0.81% among controls [202]. In research conducted by Azizi et al., the prevalence of IBD in SIgAD patients was 3.3% [37]. Both results were statistically significant.

When concerning the coexistence of SIgAD and UC or SIgAD and CD separately, the prevalence was reported as 0.63% to 7.9% (weighted average of 1.73%) and between 1.21% and 15.8% (weighted average of 2.49%), respectively, in a comprehensive review of the literature performed by Odineal and Gershwin [68]. In the aforementioned Ludvigsson cohort study, the prevalence of UC in SIgAD patients was 1.7% in contrast to 0.46% in the control group. In the same study, the prevalence of CD in SIgAD individuals was 2.4%, compared to 0.42% among controls. Both results were statistically significant, suggesting an association between SIgAD and UC, as well as between SIgAD and CD [202].

### 9.2. IBD in SIgAD: Etiology

The most commonly described pathogenetic association between SIgAD and IBD concerns primary abnormalities underlying SIgAD: A deficit of IgA, especially the secretory component. On the one hand, a lack of secretory IgA can impair the local immune mechanism in the gut and thus weaken the mucous barrier, which might translate into increased permeability of the mucosa and enhanced exposure to bowel antigens, subsequently leading to sustained inflammation within the gut mucosa [286,287]. The described mechanism might contribute to the development of autoimmunity within the GI tract, which is conclusive with the observation of an increased prevalence of autoimmune diseases in patients with SIgAD [40]. On the other hand, the deficit of secretory IgA contributes to alteration in the gut microbiota, which is also a case in IBD [56]. Interestingly, Marks et al. highlighted that GI manifestations in SIgAD patients are relatively rare compared to other primary antibody deficiencies possibly due to a compensatory increase of plasma cells producing IgM attached to a secretory component [286]. Therefore, the pathogenetic role of IgA deficiency in IBD development is uncertain and requires further investigation.

### 9.3. IBD in SIgAD: Diagnostic Difficulties, Symptoms, and Treatment

To date, the literature indicates no significant differences between IBD in SIgAD and only IBD in terms of histological features, symptoms, and treatment. Histological findings in coexistent SIgAD and UC or CD are consistent with those concerning UC or CD separately, which are described above [8,287]. Despite this, many authors underline the need of ruling out PID in patients with IBD and a consistent history of recurrent infection, failure to respond to therapy, or a particularly severe course of disease [14,15,222].

## 10. Inflammatory Bowel Disease in Common Variable Immunodeficiency

### 10.1. IBD in CVID: Epidemiology

Enteropathy is a common manifestation of CVID and might resemble CeD (discussed in detail above) or IBD [93]. It is difficult to determine the actual prevalence of the latter among patients with CVID for several reasons. Firstly, CVID is a relatively rare condition; therefore, study groups are usually small. Secondly, not many studies on this topic are available and authors usually limit themselves to stating that concomitant CVID and IBD occurrence exceeds this in the general population. 

In many reviews, the prevalence of 2–13% of IBD among CVID patients repeats [13,174,222]. Hermaszewski and Webster reported the prevalence of IBD in CVID as 4% [288]. It was approximately 6% in a study by Cunningham-Rundles et al. [289]. In a retrospective Spanish study, the estimated prevalence of IBD in patients with CVID was 3.2% [63]. However, Agarwal et al. stated that approximately 6–10% of CVID patients develop an IBD-like disorder [174]. Interestingly, a retrospective study by Daniels et al. showed that among CVID patients with GI manifestations, 35% presented with a prior diagnosis of IBD [157]. Kainulainen et al. indicated that it was a case in 1% of all CVID patients [290]. These observations lead to the proposal by Uhlig et al. of “red flags” of when to consider PID being causative for IBD, including very early onset of IBD, a very severe or refractory to conventional treatment course of the disease, the need for total parenteral nutrition, as well as suggestive family history or consanguinity, and severe infections [14].

### 10.2. IBD in CVID: Etiology

Patients with CVID are predisposed to intestinal inflammation, possibly due to T-cell dysfunctions (discussed it details above), since treatment with immunoglobulin replacement therapy does not ameliorate the course of the colitis [62,156,289,291].

The literature provides numerous studies concerning abnormalities of T-cells in CVID enteropathy; however, the results are not fully conclusive. An increased number of ILCs (CD3-) producing IFN-γ was observed in the mucous of patients with CVID but not with IBD only [93,292]. It is suggested that these might contribute to an increased Th1 response, as studies by Cols et al. and Mannon et al. showed that lamina propria T-cells of CVID patients with enteropathy resembling IBD produce increased amounts of IL−12 and IFN-γ [291,293]. The role of both cytokines in the promotion of gut inflammation in CD is well investigated [294] and therapeutic agents targeting IL−12 with briakinumab or ustekinumab have promising results [295,296,297]. However, mucosal T-cells of CVID patients produce lesser amounts of IL−17, IL−23, and TNF-α than patients with idiopathic IBD [291,293]. This suggests divergent pathogenic mechanisms underlying CVID/IBD and IBD but acting on a shared immunological background [286]. In contrast, Agarwal et al. showed that T-cells from the lamina propria of CVID patients exhibited a lower overall mRNA level for IFN-γ compared to UC and CD. This finding neared statistical significance. Furthermore, this study demonstrated impaired production of IL−2, IL−10, IFN-γ, and TNF-α in CVID/IBD patients following TCR-dependent stimulation. This is consistent with the theory described above that GI inflammation in CVID might be driven by T-cell dysfunction, including TCR impairment [174]. It is also suggested that, in rare cases, CVID and IBD can share a common genetic background since mutation of *ICOS*-*CTLA4* can underlie CVID development and might manifest as IBD [147,298].

Another cause that might underlie the development of CD in CVID patients is enhanced granulomas formation in CVID. An increased trend of monocytes forming giant cells has been demonstrated in CVID, which contributed to elevated granulocyte-macrophage colony-stimulating factor, IL−4, IFN-γ, and TNF-α [84,299]. Besides, Scott-Taylor et al. reported a greater trend of CVID monocytes fusing in immunoglobulin-conditioned media, which suggest that standard immunoglobulin replacement therapy can contribute to granuloma formation [299]. Whether this can enhance inflammation in CVID patients remains to be elucidated. Interestingly, Sanges et al. reported a case of CVID patients with coexistent CD, which were unresponsive to intravenous administration of immunoglobulins, but subcutaneous immunoglobulins turned out to be effective [300]. However, as the granulomas can be linked to an active crypt destructive colitis, the coexistence of CVID and IBD seems understandable [157].

It is established that low mucosal IgA levels can contribute to intestinal infections, luminal bacterial overgrowth, and increased permeability of the mucosal barrier, enhancing inflammation [301]. Gut microbiota alteration is observed both in CVID and IBD [144]. However, a relatively low frequency of GI manifestations in SIgAD, in contrast to other primary immunoglobulin deficiencies, strongly suggests that a decreased IgA level in neither causative nor sufficient in enteropathy development [286].

### 10.3. IBD in CVID: Diagnostic Difficulties and Symptoms

There is a lack of consensus regarding whether inflammation of the intestinal tract in CVID patients should be considered as IBD coexisting with CVID or as a distinct entity: CVID-associated enteropathy or CVID-associated colitis.

Most authors agree on that the clinical signs in IBD or IBD-like within CVID patients show no significant differences with those without CVID and include weight loss, chronic diarrhoea, rectal bleeding, abdominal pain, and malabsorption [6,62,157,160]. The main difference concerns the possibility of co-occurring severe infections in immunocompromised patients, which are included in the aforementioned “red flags” of IBD [15].

However, in terms of histological findings in CVID enteropathy/colitis, data provided by the literature are conflicting. Various histological findings are observed in CVID patients. According to Kalha and Sellin, these findings include elevated intraepithelial lymphocytosis, increased macrophages, acute inflammation in the crypt epithelium and lamina propria, or destruction of crypts, and decreased plasma cells. Granulomas and giant cells are usually not present despite an increased trend to form granulomas in CVID [302].

Hartono et al. summarized that histopathology assessment of colon specimens from CVID patients shows endoscopic and histopathological features that overlap considerably with CD or UC [157,222,303,304]. One difference that can help distinguish idiopathic IBD from CVID enteropathy/colitis remains the paucity of plasma cells in their biopsy (observed in 68% of the patients) [222,305]. Additionally, Tegtmeyer et al. indicated that both symptoms and histologic findings are indistinguishable in idiopathic IBD and IBD related to PID [15]. Similarly, Sanges et al. stated that, except a lack or paucity of plasma cells in mucosal specimens, the two discussed conditions are indistinguishable from each other based on symptoms, and endoscopic and histological examinations [300].

On the other hand, many authors are in favour of the theory that CVID enteropathy/colitis is a distinct independent disorder, separate from classic IBD, as there is an aforementioned lack of plasma cells in the mucosa [302,306,307]. Besides, this approach highlights that CVID can mimic lymphocytic colitis, collagenous colitis, and colitis associated with graft-versus-host disease [308,309,310]. According to Khodadad et al., CVID enteropathy/colitis can be divided into three major groups: Crypt-destructive colitis, non-crypt-destructive colitis, and graft versus host disease-like pattern. The former includes classic IBD, specifically UC and CD. Non-crypt-destructive colitis comprises lymphocytic colitis and collagenous colitis [306,307]. Additionally, in CVID enteropathy/colitis, stricturing is observed; however, fistulation is not common [286].

All the above-described observations suggest that some cases of gut inflammation in CVID patients can be diagnosed as true IBD, while others should be referred to as IBD-like. There is an ongoing discussion on this topic and no consensus has been reached yet.

### 10.4. IBD in CVID: Treatment

The clinical approach in managing IBD in primary antibody deficiencies remains a challenge, as the heterogeneous etiology impedes apt diagnostics, as well as treatment. Suspicion of immunodeficiency should be made, especially when patients with persistent diarrhoea, malabsorption, failure to thrive, and resistance to standard therapy present with an unusual clinical course and opportunistic or severe infections. Additionally, prior diagnosis of CVID should suggest an increased awareness concerning GI symptoms, as Resnick et al. showed increased mortality in CVID patients with GI disease and malabsorption [160].

Regardless of the doubts of whether IBD in CVID should be referred to as IBD, or rather IBD-like disease, or CVID enteropathy/colitis, most authors recommend using therapy schemes for IBD, with increased caution concerning immunosuppressive drugs [6,311]. However, there is little data about the therapeutical approach in this case. Gut inflammation in CVID is usually difficult to control and often resistant to standard IBD therapy [300,311]. Treatment of non-infectious GI disease in CVID includes corticosteroids, elimination of bacterial overgrowth with antibiotics, 5-aminosalicylic acid, 6-mercaptopurine, and azathioprine [301,311,312]. Besides, several groups reported substantial efficacy of targeted biological therapies, including anti-TNF-α drugs (infliximab and adalimumab), and the anti-IL−12/IL−23 monoclonal antibody ustekinumab [297,305,313,314,315,316,317]. Vedolizumab, an inhibitor of α4β7integrin, was also used in CVID-enteropathy, although with different results [93,318]. Furthermore, CVID patients with substantial T-cell defects require careful monitoring for fungal infections when on biological treatment.

Standard management of CVID patients includes immunoglobulin replacement therapy given intravenously or subcutaneously [86]. Immunoglobulin replacement therapy lowers the frequency of recurrent or severe infections and reduces the rate of hospitalization. In general, it was found to be ineffective in IBD-like disease in CVID [147,156,174], despite single case reports reporting the efficiency of subcutaneous immunoglobulin treatment [289,300].

In cases of severe malnutrition, patients might require total parenteral nutrition with micro- and macronutrient supplementation, electrolyte management, and prevention and treatment of fat-soluble vitamin deficiency [93,319,320]. A study conducted by Teahon et al. showed an improvement of GI symptoms in CVID/IBD patients while on an elemental diet [321].

Scarce literature available in this field, concerning the management of CVID-associated enteropathy and therapeutical outcomes, as well as potential side effects of certain therapies, provides little support when it comes to the diagnostic decision [313], hence underlying the crucial need for further investigation.

A graphical summary concerning IBD and CVID is presented in Figure 4.

## Figures and Tables

**Figure 1 ijms-21-05223-f001:**
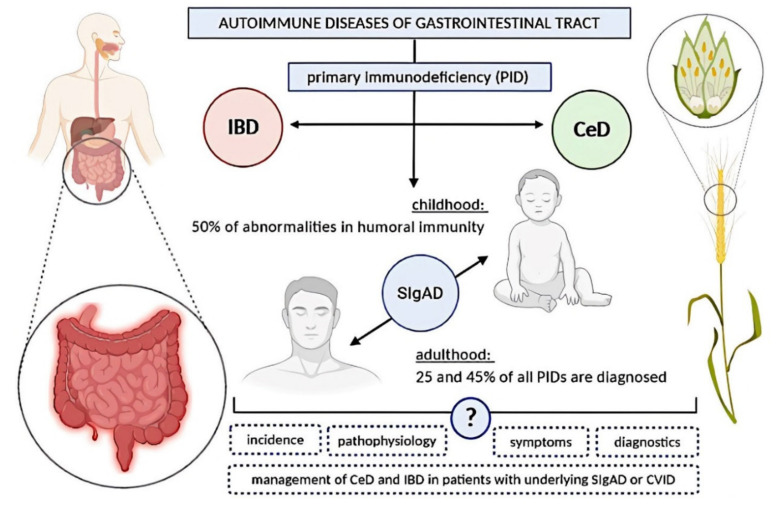
Inflammatory bowel disease and celiac disease as the most common immune-mediated gastrointestinal disorders worldwide and their association with primary immunodeficiencies. IBD, inflammatory bowel disease; CeD, celiac disease; SIgAD, selective IgA deficiency; CVID, common variable immunodeficiency.

**Figure 2 ijms-21-05223-f002:**
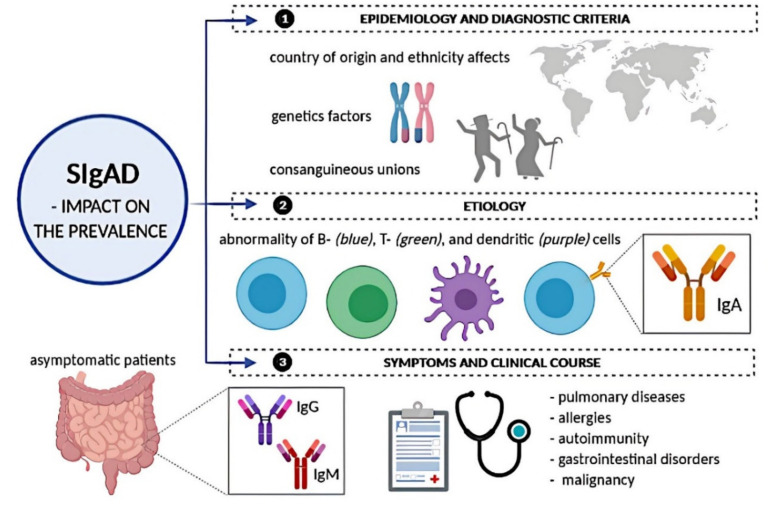
Factors affecting the prevalence of SIgAD and its aetiology and symptoms. SIgAD, selective IgA deficiency.

**Figure 3 ijms-21-05223-f003:**
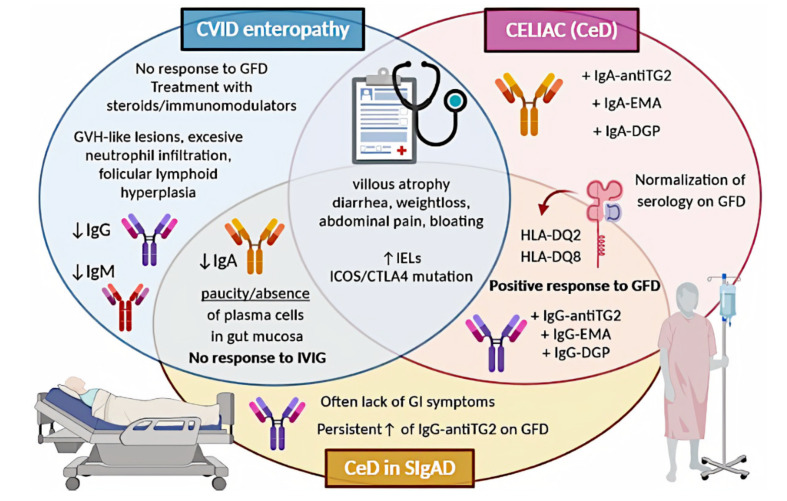
Comparrison of CeD, CeD in SIgAD, and CVID enteropathy. CVID, common variable immunodeficiency; CeD, celiac disease; SIgAD, selective IgA deficiency; GFD, gluten-free diet; GVH, graft-versus-host; IVIG, intravenous immunoglobulin; IELs, intraepithelial lymphocytes; antiTG2, anti-tissue transglutaminase; EMA, endomysial antibodies; DGP, deamidated glutin peptides; GI, gastrointestinal.

**Figure 4 ijms-21-05223-f004:**
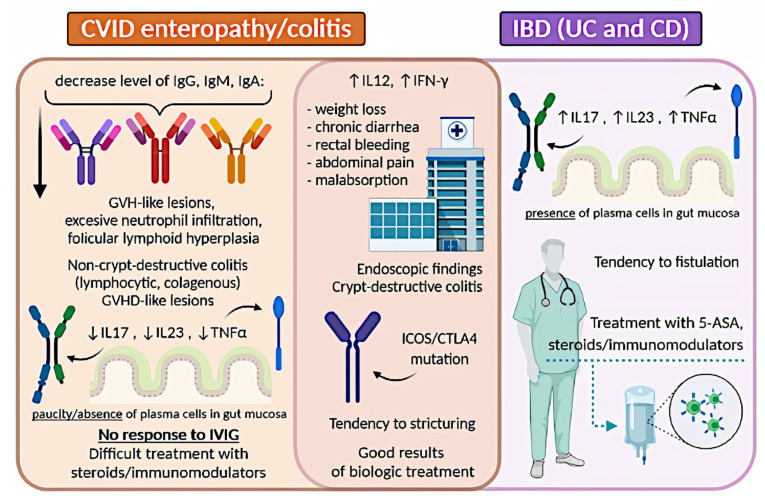
Comparrison of IBD and CVID enteropathy/colitis. CVID, common variable immunodeficiency; IBD, inflammatory bowel disease; UC, ulcerative colitis; CD, Crohn’s disease; GVH, graft-versus-host; IL, interleukin; TNF-α, tumor necrosis factor-alpha; IVIG, intravenous immunoglobulin; INF-γ, interferon-gamma; 5-ASA, 5-aminosalicylic acid.

**Table 1 ijms-21-05223-t001:** Selective IgA deficiency (SIgAD)—diagnostic criteria according to ESID (European Society for Immunodeficiencies) [23].

Selective IgA Deficiency (SIgAD)—Diagnostic Criteria	
**At least one of:**	increased susceptibility to infection
	autoimmune manifestations
	affected family member
**ALL of the following:**	serum IgA level lower than 7 mg/dL (0.07 g/L)
	normal serum levels of IgG and IgM
	patients older than 4 years
	normal IgG antibody response to vaccination
	excluded other causes of hypogammaglobulinemia
	excluded profound T-cell deficiency

**Table 2 ijms-21-05223-t002:** Common variable immunodeficiency (CVID)—diagnostic criteria according to ESID (European Society for Immunodeficiencies) [23,95].

Common Variable Immunodeficiency (CVID)—Diagnostic Criteria	
**At least one of:**	increased susceptibility to infection
	autoimmune manifestations
	granulomatous disease
	unexplained polyclonal lymphoproliferation
	affected family member with antibody deficiency
**ALL of the following:**	serum IgG and IgA levels at least 2 SD below the mean for age with/without decreased serum IgM level
	normal serum levels of IgG and IgM
	patients older than 4 years
	poor response to vaccination and/or low switched memory B-cells (<70% of age-related normal value)
	excluded other causes of hypogammaglobulinemia
	excluded profound T-cell deficiency

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
