# Peer review of "Primary Humoral Immune Deficiencies: Overlooked Mimickers of Chronic Immune-Mediated Gastrointestinal Diseases in Adults"

_ijms, 2020, doi:10.3390/ijms21155223_

Round 1

Reviewer 1 Report

Interesting review on the possibility of confusion between humoral immunodeficiencies and intestinal immunomediated pathology.
The development is interesting, well argued and wanting to be practical.
Perhaps it should be specified that reference is made to primary immunodeficiencies. As the involvement of secondary hypogammaglobulinemias (for example after using rituximab) can have different nuances, especially depending on the underlying disease.
In figure 4, INF appears to refer to interferon gamma, when you should put IFN.

Author Response

On the behalf of the authors; manuscript “Primary Humoral Immune Deficiencies - Overlooked Mimickers of Chronic, Immune-mediated Gastrointestinal Diseases in Adults”, I appreciate your helpful comments.

In our opinion, suggested changes are beneficial to the manuscript and contributed to its quality improvement. Following the guidelines, we have introduced the following changes:

Comment 1:

Perhaps it should be specified that reference is made to primary immunodeficiencies. As the involvement of secondary hypogammaglobulinemias (for example after using rituximab) can have different nuances, especially depending on the underlying disease.

Reply:

We have changed the title of the manuscript to underline that the work refers only to primary humoral immunodeficiency, as the previous title may be confusing.  Changes are marked in manuscript.

“Primary Humoral Immune Deficiencies - Overlooked Mimickers of Chronic, Immune-mediated Gastrointestinal Diseases in Adults.”

Comment 2:

In figure 4, INF appears to refer to interferon gamma, when you should put IFN.

            Reply:

We have replaced figure number 4 by a version with the corrected abbreviation for interferon gamma. This abbreviation has been corrected also in figure footage. Changes are marked in manuscript.

Reviewer 2 Report

In this review Ida JM and collaborators describe a complete review on autoimmune gastrointestinal diseases starting from occurrence, pathophysiological to diagnostics and management. The authors then compare classic forms of CeD and IBD with those observed in patients with compromised humoral immunity. In general, this is an important review that covers the whole spectrum of the GI diseases. It is very useful for experts and non-experts and is written in a way that is easy to follow even for someone who is not in the field. The figures are well presented and add more visual data -especially Fig 3 and 4- that increases the significance of this work. I believe that this will be appreciated by the IJMS readership and scientists in the field.

Couple of points to consider are:

  1. Although the manuscript is easy to follow and nicely written, the abstract reads a little complicated and can be rewritten to simplify the message of the review. 

  1. Table 1 needs a little editing, some of the words are cropped.

Author Response

On the behalf of the authors; manuscript “Primary Humoral Immune Deficiencies - Overlooked Mimickers of Chronic, Immune-mediated Gastrointestinal Diseases in Adults”, I appreciate your helpful comments.

In our opinion, suggested changes are beneficial to the manuscript and contributed to its quality improvement. Following the guidelines, we have introduced the following changes:

Comment 1:

Although the manuscript is easy to follow and nicely written, the abstract reads a little complicated and can be rewritten to simplify the message of the review. 

Reply:

We have rewritten the abstract to make it less complicated and to underline the message of the review. Changes are marked in manuscript.

“In recent years, the incidence of immune-mediated gastrointestinal disorders including celiac disease (CeD) and inflammatory bowel disease (IBD), is increasingly growing worldwide. This generates a need to identify the conditions that may compromise the diagnosis and treatment of them. It is well established that primary immunodeficiencies (PIDs) can present with gastrointestinal symptoms and mimic other diseases, including CeD and IBD. Thus PIDs may comprise CeD/IBD diagnostic processes. Although, PIDs are often considered pediatric ailments, whereas between 25 and 45% of PIDs are diagnosed in adults. The most common PIDs in adults are the selective immunoglobulin A deficiency (SIgAD) and the common variable immunodeficiency (CVID). In both conditions, a trend to autoimmunity occurs and the occurrence of CeD and IBD in SIgAD/CVID patients is significantly higher than in the general population. However, some differences concerning diagnostics and management between enteropathy/colitis in PIDs, as compared to idiopathic forms of CeD/IBD, have been described. There is an ongoing discussion whether CeD and IBD in CVID patients should be considered a true CeD and IBD or just CeD-like and IBD-like diseases. This review addresses the current state of the art of the most common primary immunodeficiencies in adults and co-occurring CeD and IBD.”

Comment 2:

Table 1 needs a little editing, some of the words are cropped.

Reply:

We found out that problem occurred due to conversion to PDF format and referred to both tables. We have edited both of them and solved the problem of cropped words. Changes are marked in manuscript.